# Histology of Cerebral Clots in Cryptogenic Stroke Varies According to the Presence of a Patent Foramen Ovale

**DOI:** 10.3390/ijms23169474

**Published:** 2022-08-22

**Authors:** Johanna Härtl, Maria Berndt, Holger Poppert, Friederike Liesche-Starnecker, Katja Steiger, Silke Wunderlich, Tobias Boeckh-Behrens, Benno Ikenberg

**Affiliations:** 1Department of Neurology, Klinikum Rechts der Isar, School of Medicine, Technical University of Munich, 81675 Munich, Germany; 2Department of Diagnostic and Interventional Neuroradiology, Klinikum Rechts der Isar, School of Medicine, Technical University of Munich, 81675 Munich, Germany; 3Department of Neurology, Helios Klinik München West, 81241 Munich, Germany; 4Department of Pathology, Klinikum Rechts der Isar, School of Medicine, Technical University of Munich, 81675 Munich, Germany

**Keywords:** ischemic stroke, patent foramen ovale, thrombus histology, mechanical thrombectomy, cryptogenic stroke

## Abstract

Although a pathophysiological impact remains difficult to prove in individual patient care, a patent foramen ovale (PFO) is currently considered of high relevance for secondary prophylaxis in selected patients with cryptogenic ischemic stroke. By quantification of histological clot composition, we aimed to enhance pathophysiological understanding of PFO attributable ischemic strokes. Retrospectively, we evaluated all cerebral clots retrieved by mechanical thrombectomy for acute ischemic stroke treatment between 2011 and 2021 at our comprehensive stroke care center. Inclusion criteria applied were cryptogenic stroke, age (≤60 years), and PFO status according to transesophageal echocardiography, resulting in a study population of 58 patients. Relative clot composition was calculated using orbit image analysis to define the ratio of main histologic components (fibrin/platelets (F/P), red blood cell count (RBC), leukocytes). Cryptogenic stroke patients with PFO (PFO+, n = 20) displayed a significantly higher percentage of RBC (0.57 ± 0.17; *p* = 0.002) and lower percentage of F/P (0.38 ± 0.15; *p* = 0.003) compared to patients without PFO (PFO–, n = 38) (RBC: 0.41 ± 0.21; F/P: 0.52 ± 0.20). In conclusion, histologic clot composition in cryptogenic stroke varies depending on the presence of a PFO. Our findings histologically support the concept that a PFO may be of pathophysiological relevance in cryptogenic ischemic stroke.

## 1. Introduction

Mechanical thrombectomy is considered the gold standard in the treatment of acute ischemic stroke due to large vessel occlusion. With the systematic retrieval of cerebral thrombi, these have become widely available for histologic analysis. Previous studies have shown a significant co-dependency of histologic clot composition and etiology of acute ischemic stroke. In particular, fibrin/platelet (F/P) and red blood cell count (RBC) vary in cardio- and arterial embolic stroke [1,2,3,4,5,6,7,8]. Despite sufficient medical clarification, the etiology of ischemic stroke remains unclear in approximately a fourth of all patients [9]. A patent foramen ovale (PFO) is significantly more prevalent in juvenile patients with cryptogenic stroke compared to healthy controls [10]. In these patients, a paradoxical (venous to arterial) or intracardiac embolic source is causatively discussed [11]. Interventional PFO closure was shown to be effective in the prevention of recurrent strokes in selected patients [12,13,14,15]. To evaluate the necessity or benefit of an interventional PFO closure in clinical decision making, age, shunt size, and shunt morphology are currently applied [12,13]. However, a validated approach to prove a causative link between cryptogenic stroke and the presence of a PFO in individual patient care is currently lacking. Contrary to available studies on cardio- and arterial embolic stroke, the impact of a PFO on histologic clot composition in patients with cryptogenic stroke has not been addressed thus far. To date, the link between PFO and stroke is based on the aforementioned epidemiologic associations, but not on histologic analysis. We therefore posed the question of whether the histologic composition of cerebral clots in patients suffering from cryptogenic stroke differs depending on the presence of a PFO. Hereby, we aimed to provide a possible marker in clinical decision making and to improve pathophysiological understanding of PFO attributable stroke.

## 2. Results

### 2.1. Patient Cohort

In total, 58 cerebral clots of patients with cryptogenic stroke aged below 60 met our inclusion criteria. Among these, 38 patients (65.5%) did not have a PFO (PFO–). In 20 patients (34.5%), a PFO was present (PFO+). In the PFO+ subgroup, a deep vein thrombosis (DVT) was diagnosed in five patients (PFO+ DVT+). DVT screening was not performed in three patients. Baseline characteristics of the study population are shown in Table 1.

### 2.2. Histological Clot Composition According to PFO Status

Comparing the PFO+ and PFO– group, PFO+ patients displayed a higher percentage of red blood cell count (RBC) (PFO– 0.41 ± 0.21; PFO+ 0.57 ± 0.17; *p* = 0.002) and a lower fibrin/platelet (F/P) count (PFO– 0.52 ± 0.20; PFO+ 0.38 ± 0.15; *p* = 0.003). Share of leucocytes did not vary significantly between both groups (PFO– 0.08 ± 0.05; PFO+ 0.06 ± 0.05; *p* = 0.1).

Boxplots of RBC, Leukocytes and F/P composition as well as a tabulation of mean values according to clot composition for each group and their calculated *p*-value are given in Figure 1A,B.

### 2.3. Histological Clot Composition in the PFO+ Patients with Respect to Proven DVT

Comparing the PFO+ patients with (n = 12) and without (n = 5) DVT, we did not detect a significant difference in histological clot composition of either RBC (PFO+DVT– 0.57 ± 0.14; PFO+DVT+ 0.64 ± 0.24; *p* = 0.46) or F/P (PFO+DVT– 0.39 ± 0.13; PFO+DVT+ 0.28 ± 0.18; *p* = 0.19). Of note, there was a distinctly elevated RBC and lower F/P share in the DVT+ subgroup (see Figure 2A,B).

Comparing the PFO+ and PFO– subgroup, the statistically significant difference of a higher RBC and lower F/P count prevailed if patients with known DVT were excluded from the PFO+ subgroup (RBC: PFO– 0.41 ± 0.21; PFO+DVT– 0.57 ± 0.14; *p* = 0.02) (F/P: PFO– 0.52 ± 0.2; PFO+DVT– 0.39 ± 0.13; *p* = 0.03).

### 2.4. ROC Analysis

To evaluate performance of histological clot analysis in the detection of a PFO itself and a possibly PFO attributable stroke, a ROC analysis for RBC and F/P was calculated.

First, we compared the PFO+ and PFO– groups. ROC-Analysis calculated an area under the curve of 0.728 for RBC and 0.712 for F/P share, respectively (see Figure 3A,B).

Postulating a pathophysiological relevance of DVT detection in PFO+ patients with cryptogenic stroke, we secondly calculated ROC analysis for RBC and F/P, comparing PFO+DVT+ and PFO– thrombi. We calculated an area under the curve of 0.774 for RBC and 0.837 for F/P respectively (see Figure 3C,D). With a sensitivity of 0.6 and a false positive rate of 0.079, an RBC of at least 0.7 indicated a quantitative histological clot composition of patients with DVT, thus suggestive for a putative paradoxical embolic stroke etiology. A possible cut-off for the F/P share in the detection of a PFO attributable stroke was calculated at 0.32, which showed a sensitivity of 0.895 and a false positive rate of 0.2.

## 3. Discussion

In summary, we present the first study evaluating the quantitative histological clot composition of cryptogenic young stroke patients according to the presence of a PFO. Our major finding is that cerebral clots of patients with cryptogenic stroke and a PFO display a significantly higher percentage of RBC and lower percentage of F/P compared to patients without PFO. Further, in PFO positive patients, the finding of a DVT was associated with a non-significant increase in RBC count. Our findings histologically support the concept that (I) a PFO has a causative role in cryptogenic ischemic stroke in selected patients and (II) based on the previously described RBC predominance in venous thrombi, these may be of venous, i.e., of paradoxical embolic, origin [16].

Cerebral clots of PFO positive patients displayed a significantly higher RBC count compared to PFO negative patients. This result is in line with a recent analysis which reported on a small subgroup of five patients [17]. However, this study focused on the immunohistology of neutrophiles and neutrophil extracellular traps in cerebral thrombi and did not discuss the comparatively small PFO subgroup. To our knowledge, no other study has investigated this topic thus far. It has been shown that histologic thrombus composition varies according to stroke etiology [4,5,7,8,18]. Thus, the observed difference in the quantitative clot composition may represent a differing stroke pathophysiology in patients with cryptogenic stroke and a PFO.

Venous thrombi are mostly considered to have a higher RBC and lower F/P fraction compared to arterial thrombi [16]. Given the higher RBC and lower F/P share in the clots of PFO positive patients compared to PFO negative patients and the reported RBC predominance in venous thrombi, our results may support the concept of paradoxical embolic stroke from venous origin in PFO positive patients.

DVT detection is currently regarded as an additional clinical feature to assess the likelihood of a PFO attributable stroke, which, however, cannot prove the impact of a PFO on stroke etiology [19]. Comparing PFO positive patients with and without DVT, the DVT positive patients showed a non-significant elevation of RBC and lower F/P content. Trying to determine an RBC and F/P cut-off in the differentiation of a possibly PFO attributable stroke, we performed a ROC analysis comparing PFO positive patients with DVT to PFO negative patients. With an area under the curve of 0.8, our data suggest that histologic analysis could be a tool to identify PFO attributable ischemic stroke. As discussed, statistical analysis was limited due to the lack of a gold standard to prove a causative role of a PFO in each patient and high frequency of DVT in all stroke patients, irrespective of PFO presence [11].

Instead of a paradoxical embolic (venous to arterial) source in PFO positive patients, an in situ intracardial thrombosis may be alternatively considered as the clot origin [11]. Mostly, the histology of cardiac emboli has thus far been described as F/P rich [4,7,8]. An embolus originating in situ from a PFO would presumably share such characteristics. Consequently, an intracardiac embolic source in PFO positive patients is not supported by our histologic data.

Similar to venous thrombi, the histological clot composition of patients with suspected large artery arteriosclerosis stroke origin (LAA) was shown to display a high RBC count [4,5,18]. A classification bias towards LAA stroke as an explanation for the presented results in our study seemed rather unlikely, for two reasons. First, each patient received an appropriate stroke work-up, with the exception of relevant atherosclerosis. Second, the baseline characteristics of the PFO+ group showed a non-significant trend regarding the absence of cardiovascular risk factors, which in turn would favor atherosclerosis. Nevertheless, the similar RBC count of LAA and PFO positive clots highlights that classification of stroke etiology is insufficient solely based on histology, but may be of additive value only.

In addition to prior pathophysiologic considerations, a high RBC and low F/P count in cryptogenic stroke patients should raise suspicion for the presence of a PFO. Matching international guidelines, which recommend PFO occlusion in cryptogenic stroke up to 60 years of age, our study inclusion criteria were chosen accordingly [20,21,22]. In addition to age, it has been shown that the size and morphology of the PFO have a relevant influence on the performance of PFO occlusion [12,13,23,24]. Shunt size, however, was not routinely assessed in our patient population due to a patient enrollment period of 10 years.

Irrespective of PFO presence, it has to be stated that conflicting results have been published which try to determine the etiology of cryptogenic stroke based on clot histology only [1,3,4,8,18,25]. Mostly, it has been found that cryptogenic stroke clots resemble cardioembolic stroke emboli [1,4,8,18]. This is in accordance to our findings in the patient cohort of PFO negative patients: histological clot composition showed a higher F/P share, indicating a possible overlap to cardioembolic cerebral clots.

In terms of relevant confounders, two considerations seem of importance in the presented patient cohort. Firstly, there was no statistically significant difference in the occurrence of posterior circulation stroke in the PFO positive and PFO negative subgroup. It has recently been shown that posterior circulation stroke presents a higher RBC count [26]. Secondly, even though there was a non-significantly elevated number of patients receiving intravenous alteplase therapy in the PFO negative subgroup, there was neither a statistically relevant difference in prior oral anticoagulation and platelet therapy nor, regarding the mean time to mechanical thrombectomy constituting two known factors of relevance, on histologic clot composition [3,7]. An impact of the differing risk profiles for cardiovascular diseases and venous thrombosis on histological clot composition has not been shown thus far.

This study is mainly limited by its retrospective and monocentric design, which implies a possible selection bias. In addition, the restricted number of patients resulted in a low power of the performed statistical tests. It has been shown that PFO attributable stroke mostly presents with small cortical ischemic lesions without vessel occlusion [27]. Relying on a large vessel occlusion for pathological clot analysis, this implies that the majority of PFO attributable strokes cannot be studied histologically. Thus, the depicted results may not be universally applicable. Additionally, the impact of prothrombotic parameters on histological clot composition is currently unclear and may present a possible confounder, which ought to be addressed by future prospective trials. The addition of immunohistochemical staining methods to selectively address minor clot components and prothrombotic factors seems relevant in future histological clot classification. Lastly, it should be kept in mind that the histologic composition of venous thrombi is not fully understood, and conflicting results have recently been published [28]. Though this may affect our pathophysiologic consideration, this does not affect our finding of a differing clot composition in cryptogenic stroke when comparing patients with and without PFO. Future histological classification of venous thrombi is required to correctly interpret a possible association of venous and arterial thrombi.

## 4. Materials and Methods

### 4.1. Study Population

Retrospectively, we extracted all cerebral clots retrieved by mechanical thrombectomy from January 2011 to March 2021 at our comprehensive stroke care center. We applied three inclusion criteria: cryptogenic stroke, age (≤60 years), and known PFO status according to contrast enhanced transesophageal echocardiography (TEE). Cryptogenic stroke was defined as an ischemic stroke without determination of an embolic source in adequate and sufficient medical clarification according to national (DGN LL) and local standard operating procedures [29]. The age cut-off was chosen to match guideline treatment recommendations for PFO closure [20,21,22]. As TEE is considered the gold standard in the detection of a PFO, no other diagnostic devices, such as transthoracic echocardiography and transcranial Doppler with bubble study, were considered mandatory in patient enrollment [30]. No other inclusion or exclusion criteria were defined. Partly, histologic parameters and baseline data were published in prior studies [8,18,26].

The study population was divided into two subgroups according to the presence or absence of a PFO: PFO negative (PFO–) and PFO positive (PFO+). The PFO+ group was further divided according to the presence or absence of a proven deep vein thrombosis (DVT), assessed by ultrasound examination. Based on the Wells score [31], ultrasound based DVT screening was performed in our cohort of PFO+ patients but was neither compulsory nor followed strict guideline recommendations. Due to a patient enrollment period of 10 years, DVT screening was not performed in all PFO+ patients included. Patients lacking DVT screening (n = 3) were excluded from the DVT subgroup analysis. The application of exclusion criteria and the study population according to PFO and DVT status are depicted in the flow chart of Figure 4.

Clinical data of stroke patients such as sex, age, site of stroke, thrombolysis, premedication, and cardiovascular risk factors including arterial hypertension, macroangiopathy, nicotine abuse, and diabetes mellitus were retrieved from medical records. Risk factors for venous thrombosis were summarized as mechanical (e.g., immobilization), medical (e.g., glucocorticoid therapy), hereditary diseases (Factor V Leiden, prothrombin gene mutation), history of prior thrombosis, laboratory diagnostics in thrombophilia (Protein S deficiency, Protein C deficiency, fibrinogen, Factor XII, Antithrombin III, APC resistance, Factor VIII), history of prior thrombosis, and comorbidities (active neoplastic diseases). Pre- and post-interventional clinical status was assessed according to the National Institutes of Health Stroke Scale (NIHSS) Score and modified Rankin Scale (mRS) by the attending neurologist at admission and discharge. Procedural neuroradiological data included time to recanalization and the thrombolysis in cerebral infarction score (mTICI), which were evaluated by a senior neuroradiologist. 

### 4.2. Standard Treatment Protocol

The diagnostic work-up of stroke etiology followed international guidelines and included at least one Holter–ECG monitoring, extra- and trans-cerebral duplex sonography, TEE, and laboratory checks determining cardiovascular risk factors such as lipid status and hyperglycemia. Stroke etiology was considered cryptogenic given the definite and sufficient exclusion of competing etiologies by the above diagnostics after evaluation by an experienced neurologist only.

### 4.3. Histologic Work-Up

For histopathologic analysis, the cerebral clots were fixed in phosphate-buffered 3.5% formalin and embedded in paraffin. Two μm slices using Microm HM 335 E microtome (Microm International GmbH, Walldorf, Germany) were cut, and hematoxylin–eosin (HE) as well as Elastica-van-Gieson (EVG) staining were performed following standard protocol and as reported before [18]. Afterwards, the slides were scanned at high resolution (400-fold) with a Leica AT2 scanner (Leica Biosystems, Wetzlar, Germany) and digitally stored.

For quantitative histologic analysis, orbit image analysis (OIA) was used (www.orbit.bio (accessed on 18 July 2022)) [32]. Performance and technical application of OIA in the histological quantification of acute ischemic stroke clots has been evaluated [32] and has been commonly used [7]. We first trained the software using a color-based machine learning technique (Figure 5). As routinely applied in histological cerebral thrombus analysis, we defined three major components: Leukocytes, RBC, and F/P [8,18]. It must be noted that, due to applied staining methods, F/P cannot be more accurately distinguished [8]. Applying the trained model, the quantitative histological tissue composition of each cerebral clot was calculated in HE and EVG staining and was finally manually checked for accuracy. In the case of clot fragmentation, all fragments were analyzed.

### 4.4. Statistical Analysis

For statistical analysis of the cerebral clots, the *t*-test was used for metric scale, normally distributed data comparing two independent subgroups. If normal distribution was lacking, the Mann–Whitney U test was calculated, respectively. For nominal scale data, the exact Fisher’s test was performed. Further, a receiver operating characteristics (ROC) analysis was implemented to identify a threshold in the RBC and F/P share comparing the PFO+ and PFO– negative group as well as the PFO+DVT+ subgroup.

All statistical analyses were performed using “R” (R version 4.1.0 (18 May 2021), R Foundation for Statistical Computing, Vienna, Austria, URL https://R-project.org/.) or SPSS (IBM Corp. Released 2021, IBM SPSS Statistics for Mac, Version 28.0.1, Armonk, NY: IBM Corp). Alpha error was set at 5%.

## 5. Conclusions

To summarize, histological clot composition in young patients suffering from cryptogenic embolic stroke varies according to the presence of a PFO: PFO positive patients display a significantly higher RBC and lower F/P count compared to PFO negative patients. Our data hence support the pathophysiological concept that a PFO constitutes a distinct risk factor for acute ischemic stroke with a possible venous to arterial paradoxical embolic stroke pathomechanism. In addition to prior epidemiologic association studies, this is the first study suggesting a pathophysiological relevance of a PFO on the etiology of ischemic stroke based on histologic tissue analysis. It has been discussed before that histopathological analyses of cerebral clots may, in the future, provide an advanced diagnostic tool in the medical clarification of stroke etiology and, consecutively, may have an impact on individualized secondary prophylactic therapy [4]. Given the small study population and retrospective design of our study, we consider the presented results as exploratory. Nevertheless, our study encourages further histologic investigation in PFO positive cryptogenic stroke. Currently, the results seem to support guideline recommendations to consider a PFO as causative for ischemic stroke in selected patients and evaluate interventional occlusion as secondary therapeutic strategy.

## Figures and Tables

**Figure 1 ijms-23-09474-f001:**
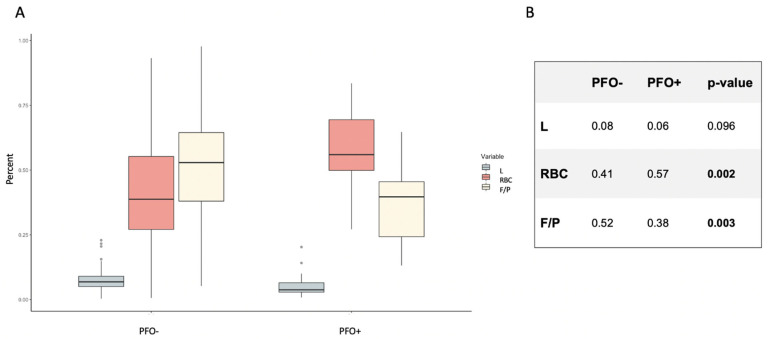
Histological clot composition in PFO– and PFO+ patients. Share of red blood cells (RBC) and fibrin/platelets (F/P) differed significantly in cryptogenic stroke depending on the presence (PFO+) or absence (PFO–) of a patent foramen ovale (PFO). (**A**) Boxplots of histological clot composition comparing PFO– and PFO+ subgroups. (**B**) Numerical overview comparing mean clot composition and the calculated *p*-value comparing PFO– and PFO+ groups. L = Leukocytes, RBC = red blood cell count, F/P = fibrin/platelets.

**Figure 2 ijms-23-09474-f002:**
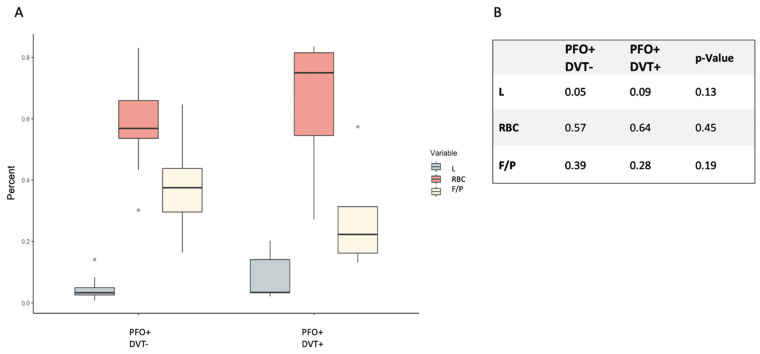
Histological thrombus composition comparing DVT+ and DVT– patients in PFO+ patients. (**A**) Boxplots of thrombus composition comparing subgroup PFO+DVT– and PFO+DVT+. (**B**) Numeric overview comparing mean thrombus composition of the PFO+DVT– and PFO+DVT+ subgroup and its calculated *p*-value. DVT = Deep vein thrombosis, PFO = patent foramen ovale, L = Leukocytes, RBC = red blood cell count, F/P = Fibrin/Platelets.

**Figure 3 ijms-23-09474-f003:**
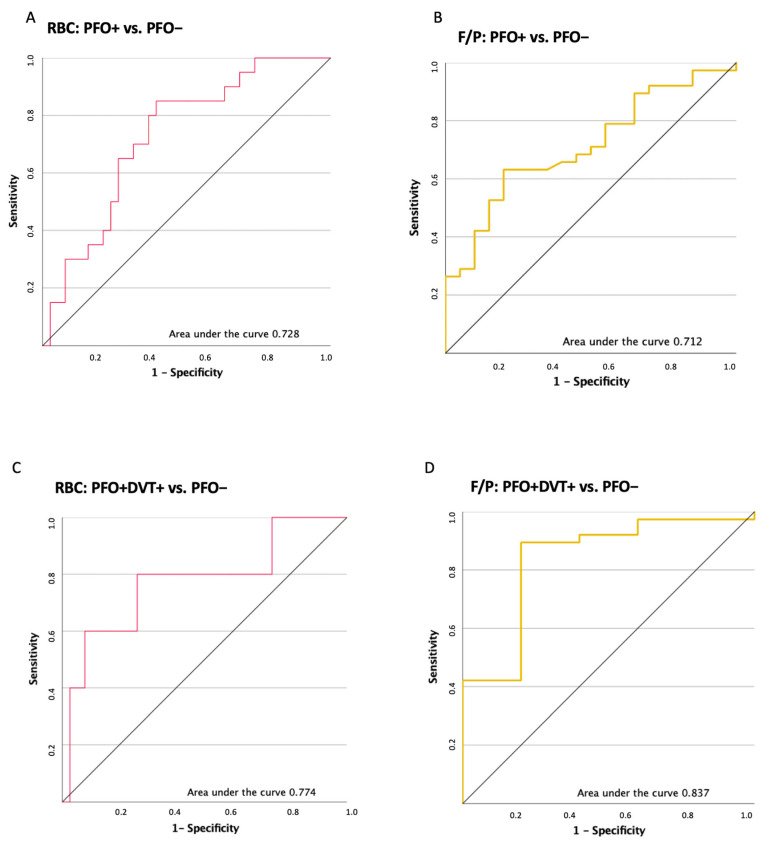
ROC analysis for red blood cell (RBC) and fibrin/platelet (F/P) share comparing PFO+ as well as PFO+DVT+ patients to PFO– patients. ROC analysis (**A**) for RBC in the PFO+ and PFO– subgroup. The area under the curve was calculated at 0.728 (**B**) for F/P share in the PFO+ and PFO– subgroup. The area under the curve was calculated at 0.712 (**C**) for RBC share in the PFO+DVT+ and PFO– subgroup. The area under the curve as calculated at 0.774 (**D**) for F/P share in the PFO+DVT+ and PFO– subgroup. The area under the curve was calculated at 0.837. RBC depicted in (**A**,**C**) in red lines. F/P share depicted in (**B**,**D**) in yellow lines.

**Figure 4 ijms-23-09474-f004:**
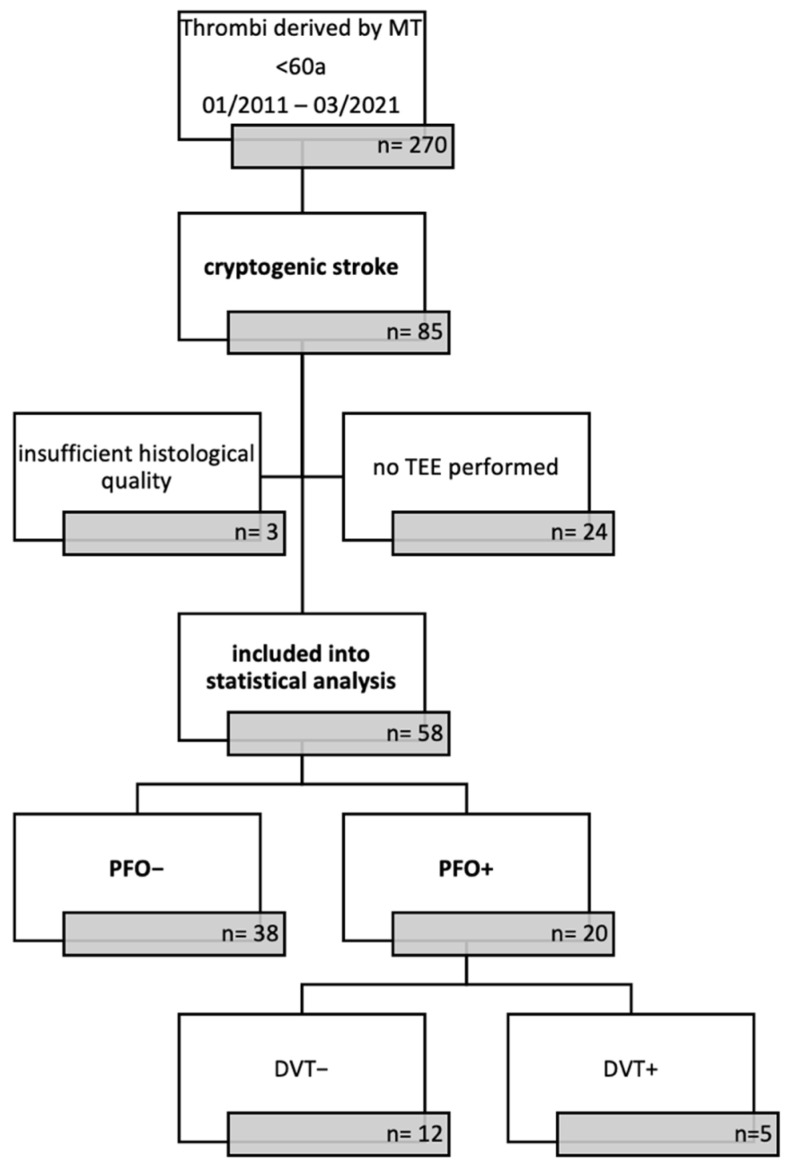
Study flow chart. Graphic depiction of patient selection criteria and division of study population. The study population was primarily divided according to the presence or absence of a patent foramen ovale (PFO+, PFO–). We secondarily divided the PFO+ patients based on the presence of a deep vein thrombosis (DVT+, DVT–), as assessed by sonography. DVT screening was not performed in three cases. a = years of age, MT = mechanical thrombectomy, TEE = transoesophageal echocardiography.

**Figure 5 ijms-23-09474-f005:**
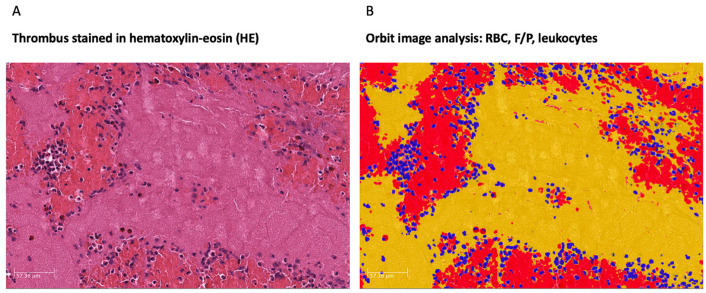
Exemplary histological clot analysis using orbit image analysis (OIA, www.orbit.bio (accessed on 18 July 2022)). (**A**) Cerebral clot in hematoxylin-eosin (HE) staining. Clots were fixed in phosphate-buffered 3.5% formalin and embedded in paraffin. Two μm slices using a Microm HM 335 E microtome (Microm International GmbH, Walldorf, Germany) were cut. HE was performed following standard protocol. The slides were scanned at high resolution (400-fold) with a Leica AT2 scanner (Leica Biosystems, Wetzlar, Germany) and digitally stored. (**B**) Cerebral clot analysis using orbit image analysis. Red blood cell (RBC) count depicted in red. Fibrin/platelet (F/P) share depicted in yellow. Leucocytes depicted in blue. Relative share of the cerebral clot was calculated by OIA.

**Table 1 ijms-23-09474-t001:** Patient characteristics. Subgroups are depicted according to the presence or absence of a patent foramen ovale (PFO+, PFO–). Patient and stroke related data including pre- and postprocedural outcome are depicted. Apt = anti platelet therapy, circ. = Circulation, MT = mechanical thrombectomy, mRS = modified Rankin Scale, mTICI = thrombolysis in cerebral infarction scale, NIHSS = National Institutes of Health Stroke Scale score, oac = oral anticoagulation, Risk for thrombosis = medical, mechanical, or laboratory increased risk of venous thrombosis or history of previous thrombosis, as outlined in the method section, * Immobilization within surgical therapy or long-distance flight, ** APC resistance, Protein C deficiency, Protein S deficiency.

	Total	PFO–	PFO+	Comparison *p*-Value
Number of patients	58	38	20	
Age *Median* *Mean* *Minimum*	47.8 50.0 21	48.7 50.0 21	50.5 45.9 22	0.68
Gender *Female* *Male*	30/51.7% 28/48.3%	20/52.6% 18/47.4%	12/60.0% 8/40.0%	0.78
Arterial hypertension	15/25.9%	13/34.2%	2/10.0%	0.06
Diabetes mellitus	6/10.3%	6/15.8%	0/0%	0.08
Regular cigarette smoking	15/25.9%	13/34.2%	2/10.0%	0.06
Risk for venous thrombosis *Glucocorticoid therapy* *Immobilization ** *Hypercoagulopathy *** *Active Malignancy*	13/22.4 %	3/7.9% 1/2.6% 2/5.3%	10/50.0% 2/10.0% 4/20.0% 3/15.0% 1/5.0%	**<0.001**
Localization of occlusion *Anterior circ.* *Posterior circ.*	53/91.4% 5/8.6%	35/92.1% 3/7.9%	18/90.0% 2/10.0%	1.0
Prior vasoactive treatment *None* *apt* *oac*	47 10 1	32 5 1	15 5 -	0.54
Median NIHSS at admission	13	14	12	0.79
Bridging Thrombolysis	36/62.0%	27/71.1%	9/45.0%	0.09
MT mTICI >2b	47/81.0%	30/78.9%	17/85.0%	0.73
Mean time to MT (min)	380	386	368	0.81
mRS at discharge *0–2* *3–6*	34/58.6% 24/41.4%	21/55.3% 17/44.7%	13/65.0% 7/35.0%	0.58

## Data Availability

Not applicable.

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
