# Peer review of "Histology of Cerebral Clots in Cryptogenic Stroke Varies According to the Presence of a Patent Foramen Ovale"

_ijms, 2022, doi:10.3390/ijms23169474_

Round 1

Reviewer 1 Report

The embolic detection was performed for patent foramen ovale in ischemic stroke rather than thrombosis.

It needs to check protein C, S, and antithrombin III, fibrin, What’s the pathway of the low F/P in this study?

The 0.7 AUC of ROC is not good, it needs more cases to increase accuracy.

Reviewer 2 Report

The manuscript from Härtl et al. presents the results obtained from the histological thrombus composition study of cryptogenic ischemic stroke patients suffering from Patent Foramen Ovale (PFO).

In spite of the restricted number of patients enrolled and the low power of the statistics tests carried out, however, the authors found the thrombus composition was different enough when comparing cryptogenic stroke in individuals with and without PFO. The higher RBC and lower F/P ratio identified in PFO+ patients with cryptogenic stroke, in comparison to those ones in the same situation but PFO-, may afford a reasonable biomarker of the stroke etiology.

Taking into account the mentioned above, this manuscript deserves to be¡ published in the IJMS.

Round 2

Reviewer 1 Report

The pathway of PFO is through coagulation pathway.

The treatment of ischemic stroke related PFO used anticoagulant agent and PFO closure.

The author explained thrombosis pathway rather than embolus pathway being out of common knowledge. (Management of Patients with Patent Foramen Ovale and Cryptogenic Stroke: An Update

Mohammad Abdelghani et al. Cardiology. 2019.)

Round 3

Reviewer 1 Report

The study design checked the cerebral clot from large vessel disease with possible thrombus formation rather than an embolic clot. It needs to check embolic RBC and F/P  from the embolic filter.

A higher proportion of hypertension, diabetes mellitus, etc. was noted in the PFO group that causes not cryptogenic ischemic stroke.

The authors did not check the exams of the coagulopathy which was a significant defect of this paper.

The small sample size causes higher RBC and lower F/P in thrombosis, it needs more cases from the correct embolic pathogen to confirm the result.